# Identification of Host Factors Interacting with a γ-Shaped RNA Element from a Plant Virus-Associated Satellite RNA

**DOI:** 10.3390/v15102039

**Published:** 2023-10-01

**Authors:** Mengjiao Li, Xiaobei Zhang, Kaiyun Huang, Zhiyou Du

**Affiliations:** College of Life Sciences and Medicine, Zhejiang Sci-Tech University, Hangzhou 310018, China

**Keywords:** cucumber mosaic virus, satellite RNA, γ-shaped RNA element, glyceraldehyde-3-phosphate dehydrogenase

## Abstract

Previously, we identified a highly conserved, γ-shaped RNA element (γRE) from satellite RNAs of cucumber mosaic virus (CMV), and we determined γRE to be structurally required for satRNA survival and the inhibition of CMV replication. It remains unknown how γRE biologically functions. In this work, pull-down assays were used to screen candidates of host factors from *Nicotiana benthamiana* plants using biotin-labeled γRE as bait. Nine host factors were found to interact specifically with γRE. Then, all of these host factors were down-regulated individually in *N. benthamiana* plants via tobacco rattle virus-induced gene silencing and tested with infection by GFP-expressing CMV (CMV-gfp) and the isolate T1 of satRNA (sat-T1). Out of nine candidates, three host factors, namely histone H3, GTPase Ran3, and eukaryotic translation initiation factor 4A, were extremely important for infection by CMV-gfp and sat-T1. Moreover, we found that cytosolic glyceraldehyde-3-phosphate dehydrogenase 2 contributed to the replication of CMV and sat-T1, but also negatively regulated CMV 2b activity. Collectively, our work provides essential clues for uncovering the mechanism by which satRNAs inhibit CMV replication.

## 1. Introduction

Satellite RNA (satRNA) is a class of subviral agents that parasitize helper viruses (HVs). Since satRNAs encode neither the RNA replicase nor the coat protein, they depend on cognate HVs for replication, encapsidation, and transmission [1,2,3,4,5]. In addition to HV replication proteins, satRNAs must acquire host components as well to fulfill efficient replication [6,7,8]. Usually, satRNAs are highly structured molecules, which provide essential RNA elements for specific interactions with viral and host proteins. Thus, the identification of viral or host factors that bind to specific RNA elements of satRNAs would deepen our understanding of molecular interactions between satRNAs and cognate HVs.

Some plant viruses have been found to be associated with satRNAs, including turnip crinkle virus (TCV), cucumber mosaic virus (CMV), Pacilliam mosaic virus (PMV), and so on [9]. One of the well-studied satRNAs is the CMV-associated satRNA, which is the second satRNA reported [10]. CMV is an economically important plant virus, infecting over 1200 plants [4]. As surveyed recently, CMV is the most dominant virus that infects vegetable crops of both *Solanaceae* and *Cucurbitaceae* in China [11]. CMV has a tripartite single-stranded RNA genome (RNAs 1–3) that encodes five viral proteins. The 1a protein encoded by RNA1 is the replication auxiliary protein, possessing methyltransferase and helicase domains. Together with 1a, the 2a protein as the viral RNA-dependent RNA polymerase (RdRP) encoded by RNA2 is responsible for CMV replication. Additionally, RNA2 encodes another functional protein, 2b, which is translated from its subgenomic RNA4A [12]. The CMV 2b protein is one of the best-studied RNA silencing suppressors; it inhibits RNA silencing by binding and inactivating siRNAs [13,14,15,16,17], reducing siRNA levels [18], or weakening the slicer activity of Argonautes 1, 4, and 5, the core components of the RNA silencing machinery [19,20,21]. In addition, the CMV 2b protein inhibits salicylic acid (SA)-induced plant defense and also dampens SA-induced autophagy [22]. CMV 2b is the key effector of viral symptom expression [23], which is mainly caused by the disruption of host microRNA functions [20,24,25]. RNA3 is a bicistronic molecule encoding the viral movement protein (MP) and coat protein (CP), both of which are required for viral movement in plants. Besides packaging of viral RNAs, the CP is also required for packing satRNAs.

A number of CMV isolates or strains are associated with satRNAs that vary in size, ranging from 330 to 405 nucleotides. Infections involving satRNAs cause profound alterations to CMV-induced disease symptoms in host plants. In a few cases, viral diseases are intensified in the presence of a satRNA isolate. For instance, sat-D4 and sat-KN cause death in tomato plants infected by CMV [10,26,27], and sat-Y causes yellowing symptoms in some *Solanaceae* species [28,29]. Very recently, Jayasinghe and colleagues [30] reported that the yellowing symptoms caused by infection with CMV and sat-Y attract CMV transmitter aphids for feeding and accelerate the formation of aphid wings. The intensification of plant diseases also occurs in plants infected with other satRNAs and cognate HVs, such TCV-associated satC [31], groundnut rosette virus-associated sat-YB3 [32], and beet black scorch virus-associated satRNA [33]. However, the majority of CMV satRNAs reduced viral titers in host plants, leading to the amelioration of viral symptoms [2,3]. This symptom amelioration is proposed to be the consequence of either the competition of satRNA with CMV RNAs for the viral replicase [34] or the interference with the suppressor activity or the decreased expression level of the CMV 2b protein [35,36]. 

Previously, we identified a highly conserved RNA element with a γ-shaped structure (γSS) in CMV satRNAs [37]. This element is referred to as γ-shaped RNA element (γRE) in this work. γRE is indispensable for satRNA survival and also required for satRNA to inhibit the replication of CMV RNA1 and RNA2 [37]. γRE is proposed to competitively bind host factors that are required for the replication of both viral genomic RNAs. Molecular interactions of CMV sat-Q with host factors have been reported previously [6,7]. In this work, pull-down assays were used to screen host protein factors from *Nicotiana benthamiana* using biotin-labeled γRE or mutated γRE (γREm) as bait. We successfully screened nine protein candidates that interact specifically with γRE. Out of these nine candidates, histone 3 (H3), small GTPase Ran3 (Ran3), and eukaryotic initiation factor 4A (eIF4A) were determined to be extremely important for infection by gfp-expressing CMV and satRNA. Moreover, we found that cytosolic glyceraldehyde-3-phosphate dehydrogenase 2 (GAPDH2) plays at least two different roles in the regulation of CMV infection. Thus, this work provides important clues for dissecting the underlying mechanism by which satRNAs reduce CMV replication.

## 2. Materials and Methods

### 2.1. Plant Materials and Viral Inoculation

*N. benthamiana* plants were grown in a plant growth room with a 16 h photoperiod and a light intensity of 150–200 µE∙m^−2^∙s^−1^ at about 22 °C. Seedlings at the 5–6-leaf stage were infiltrated with *Agrobacterium* cells harboring infectious clones of tobacco rattle virus (TRV) or CMV. The infectious clones of TRV and CMV were reported previously [37,38]. 

### 2.2. Constructed Plasmids

To down-regulate RNA transcripts of host proteins using the TRV-induced gene silencing technique [38], an approximately 300 bp DNA fragment from each of the nine host proteins was amplified using standard RT-PCR with the gene-specific primers (Appendix A). The forward and reverse primers contained a *Bam*HI and a *Sma*I at their 5′ end, respectively. PCR products were digested with both restriction endonucleases *Bam*HI and *Sma*I and inserted into the plasmid 35S:TRV2 that was pre-digested with the same endonucleases. The constructed plasmids were verified by DNA sequencing and transformed into *Agrobacterium tumefaciens* GV3101 using the CaCl_2_-mediated freeze–thaw method [39]. 

### 2.3. Pull-Down Assays

To screen γRE-binding viral and host proteins, pull-down assays were performed using commercially synthesized (Genescript Ltd., Nanjing, China) 5′ biotin-labeled γRE (wt-γRE) and its mutant (mut-γRE) as bait. One nmol of Biotin-wt-γRE or Biotin-mut-γRE was dissolved in 1X folding buffer (80 mM Tris-HCl pH 8.0, 160 mM NH_4_Cl, and 11 mM MgOAc), followed by denaturation at 95 °C for 3 min and then incubation on ice for 2 min. Each RNA sample was divided equally into two aliquots, followed by incubation with 2 mL of streptavidin agarose resins (Thermo-fisher, Rockford, IL, USA) at 4 °C overnight. Meanwhile, the same amount of streptavidin agarose resins was incubated with 0.5 mL of 1 µM biotin solution as a background control. Afterward, 35 µL of 0.5 mM biotin solution was added into each tube to block unbound streptavidin agarose resins for 30 min. After they were gently washed three times with 1X RNA washing buffer (100 mM Tris, pH 8.0, 50 mM MgOAc), the resins were incubated with 600 µL of total proteins prepared from the upper systemically infected leaves of *N. benthamiana* plants inoculated with CMV. After incubation at 4 °C for 3 h, the resins were transferred into chromatographic columns, followed by washing with 2 mL of 1× RNA binding buffer (50 mM HEPES pH 7.6, 50 mM KCl, 5 mM MgOAc, 125 mM NaCl, 2 mM DTT, 10% glycerol) containing 0.05% Triton X-100, three times. Then, 300 µL of RNA binding buffer containing 10 µg/mL RNase A was added into the columns, incubated at 37 °C for 10 min, and collected by elution. In addition, 200 µL of 1X RNA binding buffer was added to each column and collected by elution. Finally, in total, 500 µL of eluents was concentrated to approximately 40 µL using a 10K Amicon filter (Millipore, Darmstadt, Germany), and the quality of the proteins was examined via SDS-PAGE prior to the identification of protein species using LC-MS/MS as described below. Ten micrograms of each eluent sample was denatured in 1X Laemmli denaturation buffer [40] at 95 °C for 3 min, separated via 15% SDS-PAGE, and stained using the silver staining method with the SilverQuest^TM^ kit (Invitrogen, Carlsbad, CA, USA) according to the manufacturer’s instructions. 

### 2.4. LC-MS/MS

Ten-microgram samples of the protein eluents were loaded onto 15% SDS-PAGE for electrophoresis. The gel strips containing proteins were cut off from the separating gel and subjected to in-gel digestion and LC-MS/MS in the Micro Biolab (Hangzhou, China). For in-gel digestion, the gel strips were destained in 50% acetonitrile (*v*/*v*) with 50 mM NH_4_HCO_3_, dehydrated with 100% acetonitrile, and rehydrated with 10 µg/µL trypsin in 50 mM NH_4_HCO_3_ and 5 mM CaCl_2_ on ice for 10 min, followed by digestion at 37 °C overnight. The tryptic peptides were extracted from the gel strips with 50% acetonitrile and 5% formic acid, followed by 100% acetonitrile. The extracted peptides were dried completely and re-suspended in 2% acetonitrile and 0.1% formic acid.

The tryptic peptides were loaded onto a homemade reverse-phase analytical column (15 cm in length, 75 µm in diameter) with a constant flow rate of 300 nL/min on an EASY-nLC 1000 UPLC system. The peptides were subjected to an NSI source followed by tandem mass spectrometry (MS/MS) in a Thermo Scientific Orbitrap HF coupled with the UPLC. The resulting MS/MS data were processed using Maxquant, and tandem mass spectra were searched against Uniprots of *N. benthamiana* and CMV. 

### 2.5. Agroinfiltration Assays in N. benthamiana Plants

*Agrobacterium* cells were cultured in Luria-Bertani liquid media containing 50 ng/mL kanamycin and 25 µg/mL rifampin at 28 °C. The cells carrying 35S:TRV1 were equally mixed with those carrying each 35S:TRV2 variant and infiltrated into the fifth leaves of approximately 3–4-week-old *N. benthamiana* plants. When the plants inoculated with the bacteria cells carrying 35S:TRV1 and 35S:TRV2-NbSu (TRV2 carrying a portion of the endogenous *Su* gene) displayed extensive yellowing in the upper systemic leaves, the upper systemic leaves of the plants infiltrated with the cells harboring 35S:TRV1 and 35S:TRV2 variants were infiltrated with the *Agrobacterium* cells harboring the infectious clone of pCb301-F109 for Fny-CMV RNA1, pCb301-F209 for Fny-CMV RNA2 and pCb301-F309-gfp for Fny-CMV RNA3 expressing GFP, or pCb301-sat-T1 for the isolate T1 of satRNA (sat-T1) as described previously [37,41]. At 3 days post-infiltration (dpi), the infiltrated leaves were photographed under UV light and collected for immunoblots and RNA gel blots. To test the contributions of host proteins to the replication of CMV and satRNA, we conducted *trans*-replication assays as described previously [37,41]. Briefly, a mixture of *Agrobacterium* cells was infiltrated into the leaves of the TRV-inoculated *N. benthamiana* plants as aforementioned, to transiently co-express CMV replicase components (1a and 2a) and the RNA silencing suppressor p19 encoded by tomato bushy stunt virus (TBSV), together with CMV RNA1Δ1a lacking 1a expression, RNA2Δ2a lacking 2a expression, RNA2Δ2a2b expressing neither 2a nor 2b, wild-type RNA3, or sat-T1. At 3 dpi, total RNAs were extracted from the infiltrated leaf tissues and subjected to RNA gel blotting analyses as described below.

### 2.6. Reverse Transcriptase Quantitative Polymerase Chain Reaction (RT-qPCR)

To measure the efficiency of TRV-induced gene silencing in *N. benthamiana* plants, RT-qPCR was carried out according to a procedure described previously [42]. Briefly, total RNAs were extracted from the upper systemic leaves of the TRV-inoculated plants and digested with Turbo DNase (Ambion, Carlsbad, CA, USA) to remove cellular DNAs according to the manufacturer’s instructions. The DNA-free RNA samples were used as templates for the synthesis of the first-strand complementary DNAs (cDNAs) using reverse transcriptase Superscript III (Invitrogen) and random hexamer primers (Takara, Dalian, China). The resultant cDNAs were diluted 10 times and used as templates for quantitative PCR reactions using 2 × SYBR-containing PCR mixture (ABI) and the gene-specific primers (Appendix A). qPCR data were analyzed using LinRegPCR software [43], and fold changes in transcript abundance were calculated using ΔΔCt methodology [44]. 

### 2.7. Immunoblot

Immunoblotting assays for the detection of GFP or CMV CP were performed as described previously [13]. Briefly, total proteins were extracted from the infiltrated patches of the leaves using phosphate-buffered saline (0.14 M NaCl, 0.01 M potassium phosphate, pH 7.4). Total proteins were separated on 15% SDS-PAGE gels and transferred onto nitrocellulose membranes (GE). Subsequently, the membranes were blocked with 1X TBS solution containing 5% fat-free milk powder at room temperature for 2 h and blotted using serum anti-GFP (Santa Cruz, Dallas, TX, USA) or homemade polyclonal antibody against CMV CP at room temperature for 2 h. The first antibody was detected using horseradish peroxidase (HRP)-conjugated anti-rabbit IgG (Abcam, Waltham, MA, USA) and an enhanced chemiluminescence solution (Thermo-Fisher). 

### 2.8. RNA Gel Blot

RNA blotting assays for the detection of CMV RNAs and satRNAs were carried out as described previously [37]. Briefly, total RNAs were extracted from the infiltrated patches of the leaves using RNA extraction buffer (0.05 M NaOAc pH 5.2, 0.01 M EDTA pH 8.0, and 1% SDS) and separated in 1.5% agarose gels containing 7% formaldehyde. Total RNAs were transferred onto positively charged nylon membranes (Amersham, Little Chalfont, UK). The digoxin (DIG)-labeled DNA oligonucleotide probes used for the detection of CMV RNAs or satRNAs have been described previously [13,37]. The DIG-labeled probes were detected using the DIG High Prime DNA Labeling and Detection Starter Kit II (Roche, Mannheim, Germany). 

## 3. Results

### 3.1. Disruption of the γRE Structure Abolished satRNA Viability in Plants

γRE is a three-way branched RNA element that contains a pseudoknot (PK1) formed between the loop sequence of the top-left hairpin (tlH) and its upstream stretch sequence (Figure 1a). PK1 is completely conserved in all CMV-associated satRNAs reported and is essential for satRNA survival in plants [37]. The top-right hairpin (trH) in γRE is required for satRNA to inhibit CMV accumulation [37]. Here, we introduced mutations into the γRE element to disrupt both PK1 and trH in the background of sat-T1 (Figure 1a) and tested its viability in the presence of its helper virus Fny-CMV in *N. benthamiana* plants. Northern blotting analyses showed that CMV RNA3 was detected markedly in local leaves inoculated with CMV and either wild-type sat-T1 or its γRE mutant (T1-γREm) (Figure 1b). As expected, sat-T1 was detected substantially, and T1-γREm was undetectable in three independent leaf samples (Figure 1b), demonstrating that the mutated γRE is inactive in supporting sat-T1 replication. This is consistent with our previous work [37]. 

### 3.2. Identification of the γRE-Binding Host Proteins

To determine γRE-interacting proteins, biotin-labeled wild-type γRE (wt-γRE) and the mutated γRE (mut-γRE) were used as bait to pull down proteins from the leaves of CMV-infected *N. benthamiana* plants (Figure 2a). The sequence of γRE or its mutant ranges from the positions 181 to 248 nt of sat-T1. In parallel, an RNA-free control was included in this assay. The proteins obtained from this assay were separated via a 15% SDS-PAGE gel, followed by silver staining (Figure 2b). The band pattern of the control sample was highly similar to that of either wt-γRE or mut-γRE, indicating that no difference between treatments and control likely was due to the abundance of proteins interacting with γRE that were below the limits of detection for silver staining. 

Five protein samples, as shown in Figure 2b, were subjected to LC-MS/MS. The data showed that a total of 139 unique peptides were identified in these samples, among which 27 were commonly present (Figure 2c). Eleven unique peptides were present in both wt-γRE samples (Figure 2c). Searching proteins for these 11 peptides showed that two of them corresponded to CMV CP and the remaining matched with nine different proteins of *N. benthamiana* (Table 1). Cytosolic glyceraldehyde-3-phosphate dehydrogenase 2 (GAPDH2) and PDR-type ABC transporter (PDR9) were commonly present in both wt-γRE samples. Other host factors include two histone proteins, H2b and H3; eIF4A; and Ran3, all of which have the ability to bind nucleotides or nucleic acids.

### 3.3. Biological Relevance of the γRE-Binding Host Factors in Infection by CMV and satRNA

To determine how relevant these γRE-binding host proteins are, we employed a TRV-mediated gene silencing technique to knock down the mRNA level of each host protein in *N. benthamiana* plants. These nine host proteins were randomly divided into two groups for testing. The mRNA transcripts for each protein tested were knocked down by 70–80% in the upper systemic leaves of the TRV-inoculated plants at 10 dpi (Appendix A). Down-regulation of these genes caused diverse impacts on plant development (Appendix A). Down-regulation of NbH2b or ubiquitin (NbUb) was detrimental to plant survival, leading to plant death at about 14 dpi. Thus, both NbH2b and NbUb were excluded from further investigation. The plants with down-regulated NbRan3, NbeIF4A, NbH3, or PDR9 showed moderate influences on plant phenotypes, including leaf distortion and yellowing. Mild alterations to plant phenotypes were observed for the plants with down-regulated NbGAPDH2, NbPAP2, or NbZAR1. 

Then, we tested co-infection with sat-T1 and the GFP-expressing CMV variant (CMV-gfp) in the upper systemic leaves of these plants with down-regulated host mRNAs. GFP fluorescence showed that the leaves with down-regulated NbPDR9, NbGAPDH2, NbPAP2, or NbZAR1 displayed a level of GFP fluorescence strength similar to that produced from the TRV-mCherry control, while the leaves with down-regulated NbH3, NbRan3, or NbeIF4A showed extremely weak or invisible GFP fluorescence (Figure 3a). In general, the GFP fluorescence observed was consistent with the relative levels of GFP and the GFP-expressing RNA3 in these leaves (Figure 3b). These results indicated that the down-regulation of NbH3, NbRan3, or NbeIF4A was detrimental to CMV infection, resulting in the poor accumulation of sat-T1 (Figure 3b). Although the down-regulation of NbPDR9, NbGAPDH2, NbPAP2, or NbZAR1 had no or limited effects on CMV RNAs, it altered the level of sat-T1 diversely. The down-regulation of NbPDR9 increased the level of sat-T1 by 23%, while the down-regulation of the other three genes reduced the level of sat-T1 (13–45%). These results suggested that these four host proteins may regulate the replication of sat-T1 in plants. We also cannot rule out the possibility that NbH3, NbRan3, or NbeIF4A is directly involved in the replication of sat-T1.

### 3.4. Contribution of GAPDH2 to the Accumulation Level of CMV RNAs and sat-T1 in Trans-Replication Assays

GAPDH2 was reported previously to be essential for CMV infection in *Arabidopsis thaliana* and proposed to be engaged in the interaction with CMV replicases [7]. However, we did not observe a marked effect of down-regulating GAPDH2 on the accumulation of CMV-gfp and sat-T1 in *N. benthamiana* plants (Figure 3). 

To clarify the biological significance of GAPDH2 in the replication of CMV and satRNA in *N. benthamiana* plants, we employed a trans-CMV-replication system [37] to determine the replication of CMV RNA1Δ1a (lacking the expression of the 1a protein), RNA2Δ2a (lacking 2a expression), RNA2Δ2a2b (expressing neither 2a nor 2b), RNA3, or sat-T1 individually in the GAPDH2-silenced leaves of *N. benthamiana*. Northern blotting analyses showed that down-regulation of NbGAPDH2 reduced the replication of RNA1Δ1a by 74% and had moderate effects on the replication of RNA3 and sat-T1, with a reduction of 37% and 22%, respectively (Figure 4). Unexpectedly, GAPDH2 silencing had a positive effect on the accumulation levels of RNA2Δ2a and its subgenomic RNA4A, with an approximately 2- to 3-times increase, which is opposite to the effect on RNA1Δ1a, RNA3, or sat-T1. Regarding RNA2Δ2a which has the capability of producing the 2b protein, we wondered whether the positive effect was due to the presence of the 2b protein. To answer the question, we tested another RNA2 variant RNA2Δ2a2b that expresses neither 2a nor 2b. Northern blotting analyses showed that although RNA2Δ2a2b and its subgenomic RNA4A were undetected in the #3 plant of the TRV-mCherry control, it accumulated substantially in the other two plants (#1# and #2), where it was about 2 times higher than that in the GAPDH2-silenced samples. This result indicates that the replication of RNA2Δ2a2b is positively regulated GAPDH2. Collectively, our data demonstrate that GAPDH2 is indeed required for not only CMV replication but also for satRNA replication and imply that GAPDH2 plausibly negatively regulates the 2b-mediated stability of CMV RNA2. 

### 3.5. GAPDH2 Negatively Regulates the Accumulation of CMV in N. benthamiana

The finding that CMV 2b influenced GAPDH2’s regulation of the accumulation of RNA2 prompted us to test the accumulation levels of CMV and its 2b-deletion mutant CMVΔ2b in GAPDH2-silenced or control plants. As mentioned earlier, GAPDH2 was down-regulated in *N. benthamiana* plants by inoculating them with TRV-GAPDH2. Ten days later, CMV and CMVΔ2b were separately inoculated via agroinfiltration on the upper leaves of the GACP2-silenced or control (TRV-mCherry) plants, and they were detected in the infiltrated leaves at 3 dpi using RNA gel blots and immunoblots. In the case of CMV, the down-regulation of GAPDH2 increased the accumulation levels of viral RNAs by over 78% and the CP by 178%, compared with the control (TRV-mCherry) (Figure 5a). These data confirm the important role of 2b in GAPDH2’s regulation of CMV accumulation. However, we did not observe a discernable difference in the accumulation levels of CMVΔ2b between the GAPDH2-silenced and the control plants (Figure 5b), suggesting that the unsilenced amounts of GAPDH2 or its homologs (GAPDH1 and GAPDH3) could be sufficient for the replication of CMVΔ2b in the GAPDH2-silenced plants. 

To determine the role of the NbGAPDH2 homologs (NbGAPDH1 and NbGAPDH3) in CMV accumulation, we silenced these three GAPDHs (NbGAPDHs) simultaneously by infiltrating *N. benthamaina* plants with *Agrobacterium* cell mixtures carrying TRV2-NbGAPDH1, TRV2-NbGAPDH2, and TRV2-NbGAPDH3 together with TRV1. Meanwhile, agroinfiltration with TRV-mCherry was included as the control. The simultaneous silencing of these three GAPC homologs had no discernable effect on plant phenotype after 10 days (Appendix A). Then, we tested the accumulation of CMV-gfp and sat-T1 together in the NbGAPDH-silenced and the control plants. At 3 dpi, the infiltrated patches of the NbGAPDH-silenced plants had discernibly stronger GFP fluorescence than those of the control plants (Figure 6a), which is consistent with the difference in the levels of viral RNAs (Figure 6b). Likewise, the accumulation level of sat-T1 increased by 37% in the NbGAPDH-silenced plants, compared with the control plants (Figure 6b).

## 4. Discussion

Following our previous work that identified the biologically important γRE in CMV satRNAs, we successfully identified nine γRE-binding host proteins in this study. Of these nine, seven host proteins were evaluated for their roles in the replication of satRNA and its HV. Our data indicate that three host factors (Ran3, eIF4A, H3) play extremely important roles in infection by CMV-gfp and sat-T1. Although GAPDH2 silencing has no obvious influence on the accumulation levels of CMV-gfp or sat-T1, it indeed reduces the accumulation level of the non-coding RNA1 (RNA1Δ1a), the non-coding RNA2 (RNA2Δ2a2b), RNA3, or sat-T1 in the *trans*-replication system. Interestingly, GAPDH2 silencing increases the accumulation of RNA2Δ2a, which expresses 2b but not 2a, which is consistent with the data showing that GAPDH2 silencing increased the accumulation of wild-type CMV but not CMVΔ2b. This work paved the way to uncover the molecular mechanism by which satRNAs attenuate CMV pathogenicity via the key element γRE. 

As a ubiquitous enzyme involved in the glycolysis pathway, GAPDH has been reported to be closely associated with the replication of some viruses [7,45,46]. Our data obtained from the *trans*-replication assay suggest that GAPDH2 plays a role in the replication of not only CMV RNAs, but also sat-T1 (Figure 4). This is consistent with the previous report that CMV fails to infect GAPDH2-knockout *Arabidopsis* mutants [6]. Chaturvedi and colleagues [6] found that GAPDH2 interacts with CMV replication proteins 1a and 2a and is required for the 1a–2a interaction, which explains the biological relevance of GAPDH2 in CMV infection. Unexpectedly, we found that the down-regulation of GAPDH2 increased the accumulation level of RNA2Δ2a in the *trans*-replication assay (Figure 4b). This is in agreement with the data showing that GACP2 silencing increased the accumulation of wild-type CMV, but not CMVΔ2b (Figure 5). Both data suggest that the CMV 2b protein contributes to viral accumulation under the circumstances of the reduced level of GAPDH2 in host cells. CMV 2b is a well-known RNA silencing suppressor protecting viral RNAs from degradation by host siRNA-mediated antiviral silencing, and it is targeted by tobacco calmodulin-like protein (rgs-CaM) for degradation, probably via autophagy [47]. Very recently, Shukla et al. [22] reported that host autophagy is induced by CMV infection and reduces viral accumulation but is compromised by the CMV 2b protein. The down-regulation of GAPCs would enhance host autophagy since GAPCs function as an inhibitor of autophagy by interacting with the autophagy protein ATG3 [48]. Thus, in the circumstance of silencing GAPDH2, the increased accumulation of CMV RNAs promoted by the expression of the CMV 2b protein could be the outcome of dampening the enhanced autophagy.

Previously, Chaturvedi and Rao [7] screened the chloroplast-located GAPDH isoform (GAPA) using positive-sense sat-Q as bait in in vitro pull-down assays. Here, we identified GAPDH2 that binds to sat-T1 γRE, but not the structurally disrupted mutant γREm (Figure 2), suggesting that either the structure of γRE or the mutated nucleotides are required for GAPDH2 binding. Cytoplasmic GAPDH has been characterized as an RNA-binding protein that specifically binds to AU-rich elements [49]. Interestingly, sat-T1 γRE contains an AU-rich sequence (223-AAAUUUCGAAAGAAA-237) (Figure 1a), leading us to speculate that the AU-rich sequence could be the GAPDH2 binding site. The most important point is the potential biological relevance of the GAPDH2-γRE binding. Wang and Nagy [46] identified the interaction of the yeast GAPDH Tdh2p with the AU pentamer sequence in the negative-strand (−) RNA of TBSV and determined that the interaction promotes the synthesis of the positive-strand (+) RNA by retaining the (−) RNA as replication templates. In contrast to the HV that has an accumulation level of (+) RNAs 100-fold greater than that of (−) RNAs in infected leaves [50], CMV satRNA accumulates its (+) RNAs and (−) RNAs at the same order of magnitude [51]. Thus, it is possible that GAPDH2 binding to γRE increases the synthesis of satRNA (−) RNA, which leads to comparable levels of satRNA (+) RNAs and (−) RNAs in infected cells. As we proposed previously, sat-RNAs inhibit the replication of CMV RNA1 and RNA2 by competing for host factors with both viral genomes in infected cells [37]. Thus, GAPDH2 could be one of the host factors competed by satRNAs and viral genomic RNAs during their replication. 

Our data demonstrate that three host factors, namely eIF4A, Ran3, and H3, are extremely important for CMV infection (Figure 2). These factors could be required for viral translation, replication, or both. eIF4A is a component of the eIF4F complex that is responsible for mRNA translation. Application of the eIF4A chemical inhibitor PatA arrests the biosynthesis of viral proteins, leading to the failure of influenza A virus replication [52]. Thus, the dramatic inhibition of CMV infection by silencing eIF4A could be the consequence of the poor translation of viral RNAs in the initial step of infection. Some studies have demonstrated that translation elongation factor 1A (eEF1A) functions as a cofactor of the viral replication complex by interacting with viral replicases or viral RNAs [53,54,55]. Thus, it is possible that eIF1A is engaged in CMV replication. Regarding the molecular function of eIF4A as an RNA helicase, it would be interesting to determine whether eIF4A binding to γRE disrupts the sat-T1 structure to negatively regulate sat-T1 infection. The small GTPase superfamily includes five subfamilies, Ras, Rho, ARF, Rab, and Ran [56]. Of these, Rab GTPase has been studied well in virus infections, including viral replication, vesicle trafficking, and endomembrane sorting [57,58,59]. Here, we found that Ran3 GTPase is critical for CMV infection (Figure 2). Such an important role of Ran3 was not yet reported in virus infection. Similarly, histone 3 (H3) also plays important roles in CMV infection. Regarding the importance of eIF4A, Ran3, and H3 in CMV infection, we speculate that sat-RNA might inhibit CMV infection by sequestering one or more of them via γRE in plant cells. 

In summary, we successfully identified nine γRE-binding host proteins, among which eIF4A, Ran3, and H3 are crucial for CMV infection. Meanwhile, GAPDH2 was found to be a positive regulator in the replication of CMV RNAs or satRNAs, but it negatively regulates infection by CMV and sat-T1 in the presence of the CMV 2b protein. This work provides primary clues for uncovering the mechanism by which satRNAs inhibit CMV replication.

## Figures and Tables

**Figure 1 viruses-15-02039-f001:**
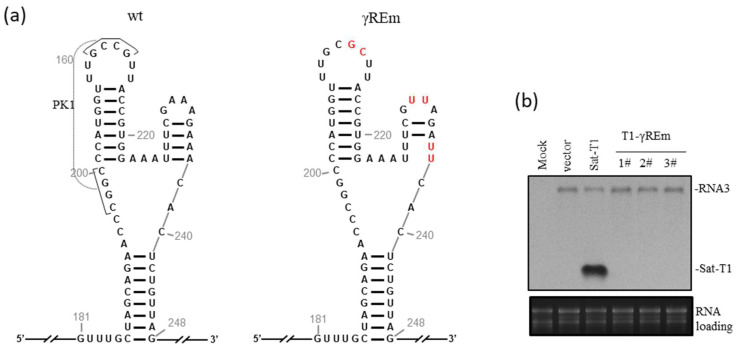
Disruption of γRE structure is lethal to sat-T1 in CMV-infected plants. (**a**) The schematic diagrams of sat-T1 and its mutant (T1-γREm) showing wild-type γRE (wt) and its mutant (γREm). The nucleotides mutated in γRE are colored red. The pseudoknot structure (PK1) and the top-right hairpin would be disrupted due to the introduced mutations. The remaining sequences in sat-T1 and its mutant are shown by double slashes. (**b**) Northern blotting analyses of CMV RNA3 and sat-T1 WT or its mutant T1-γREm. Three biological samples (1–3#) of T1-γREm were used for analysis of its viability. Total RNAs were isolated from the inoculated leaves of *N. benthamiana* at 3 days post-infiltration (dpi). CMV RNA3 and sat-T1 were detected simultaneously using DIG-labeled DNA oligonucleotides. Ribosomal RNAs were used to assess relative loading amounts.

**Figure 2 viruses-15-02039-f002:**
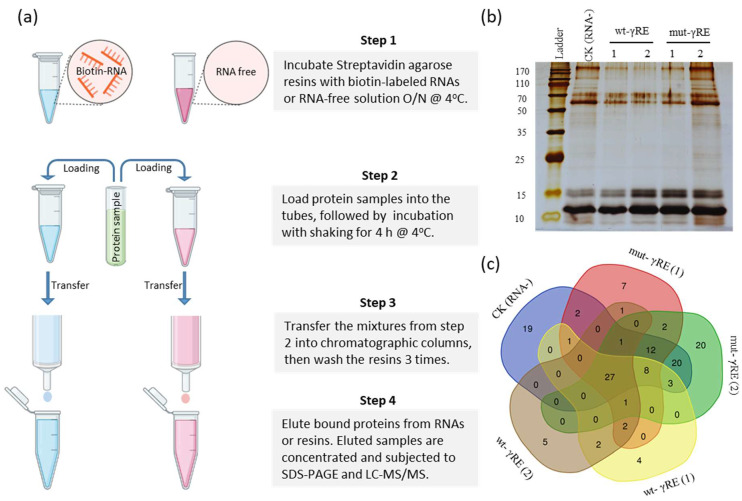
Identification of host factors pulled down by biotin-labeled wt-γRE or mut-γRE in vitro. (**a**) Schematic diagram of these 4 key steps in the pull-down assays. Biotin-labeled wt-γRE or mut-γRE was used as bait to pull down proteins. Meanwhile, an RNA-free control was included in this assay. (**b**) The protein samples separated in SDS-PAGE were stained with silver nitrate. The RNA-free control sample is the eluent from the streptavidin resin incubated only with total proteins. Two independent protein samples were pulled down by wt-γRE or mut-γRE. Total proteins were prepared from the CMV-infected leaf tissues of *N. benthamiana*. (**c**) Venn diagram of 139 unique peptides identified from these five protein samples by LC-MS/MS.

**Figure 3 viruses-15-02039-f003:**
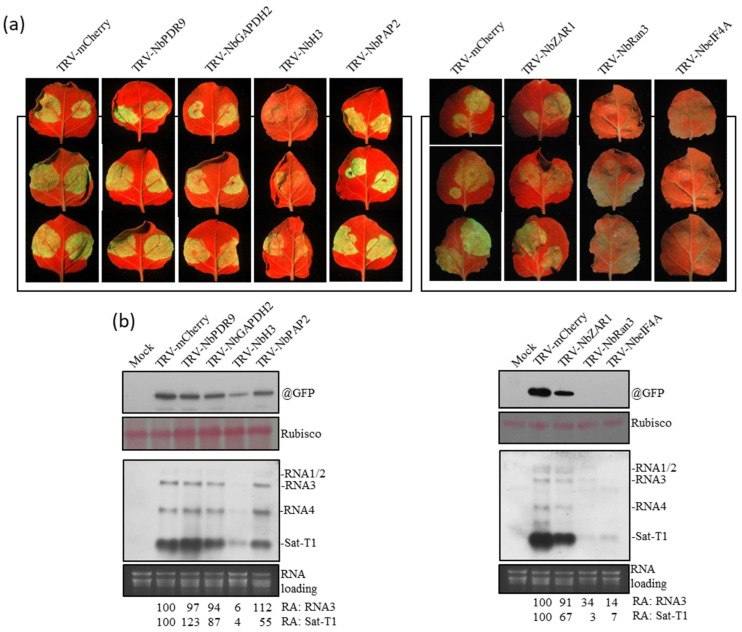
Down-regulation of host factors had distinct influences on the replication of CMV-gfp and sat-T1 in *N. benthamiana* plants. (**a**) GFP fluorescence on leaves inoculated with CMV-gfp and sat-T1 together after TRV-induced RNA silencing for target genes as shown at the top. Leaves were photographed under UV light at 3 days post-inoculation (dpi). (**b**) Molecular detection of GFP and CMV with sat-T1 using immunoblot and RNA gel blot, respectively. Both protein samples and RNA samples were prepared from the mixed leaves as shown in (**a**) at 3 dpi. Rubisco protein and ribosomal RNAs (rRNAs) were used as loading assessments for immunoblot and RNA gel blot, respectively. Relative accumulation (RA) values of RNA3 and sat-T1 are shown at the bottom. The RA values of RNA3 and sat-T1 are defined as 1 in the TRV-mcherry-treated leaves.

**Figure 4 viruses-15-02039-f004:**
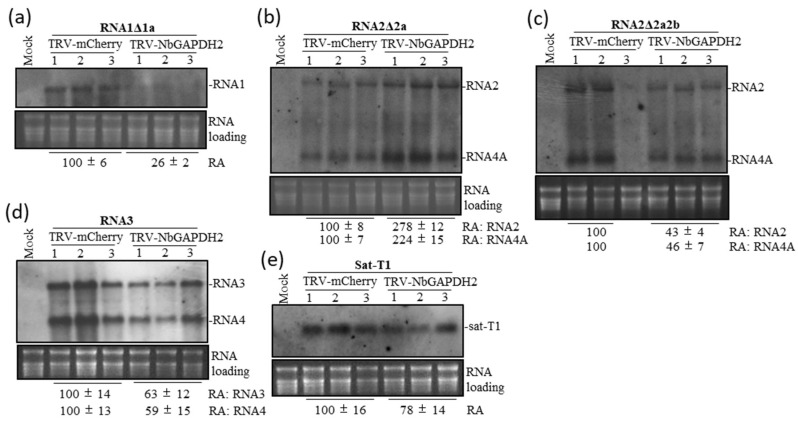
GAPDH2 positively regulated the replication of CMV RNAs and sat-T1 promoted by the transiently expressed CMV replicases in *N. benthamiana* plants. *N. benthamiana* plant leaves were primarily inoculated with either TRV-mCherry or TRV-NbGAPDH2 via agroinfiltration. Ten days later, the upper systemic leaves of the TRV-infected plants were infiltrated with *Agrobacterium* cells to co-express the CMV 1a and 2a and TBSV p19 proteins and one of the following RNAs: RNA1Δ1a (lacking the 1a protein) (**a**), RNA2Δ2a (lacking the 2a protein) (**b**), RNA2Δ2a2b (lacking the 2a and 2b proteins) (**c**), RNA3 (**d**), and sat-T1 (**e**). (**a**–**e**) Northern blotting analyses of the accumulation levels of viral RNAs or sat-T1 in the infiltrated leaves. Total RNAs were extracted from the infiltrated leaves at 3 dpi. Three biological repeats were conducted for each treatment. Relative accumulation levels of each RNA tested are shown at the bottom.

**Figure 5 viruses-15-02039-f005:**
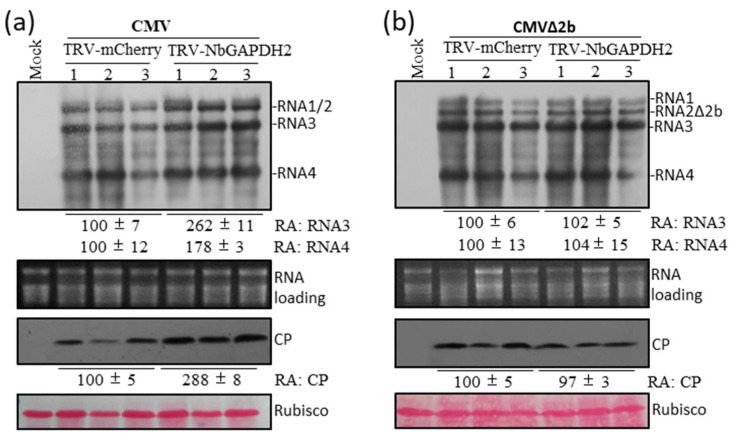
Down-regulation of GAPDH2 enhanced the accumulation of wild-type CMV but not CMVΔ2b in *N. benthamiana* plants. Plants at 3–4 weeks old were infiltrated with *Agrobacterium* cells for inoculation with TRV-mCherry or TRV-NbGAPDH2. At 10 dpi, the upper leaves were infiltrated with *Agrobacterium* cells for inoculation with wild-type CMV (**a**) or its mutant CMVΔ2b (lacking the 2b gene) (**b**). Total RNAs were extracted from the infiltrated leaves infected by CMV or CMVΔ2b at 3 dpi and tested for the accumulation levels of viral RNAs using Northern blot analyses. Total proteins were prepared from the infiltrated leaves at 3 dpi as well and tested for the accumulation of CMV CP using Western blot analyses. Ribosomal RNAs and large rubisco subunit were used as loading assessments for Northern and Western blot analyses, respectively. Relative accumulation (RA) values of RNA3 and CP are shown at the bottom.

**Figure 6 viruses-15-02039-f006:**
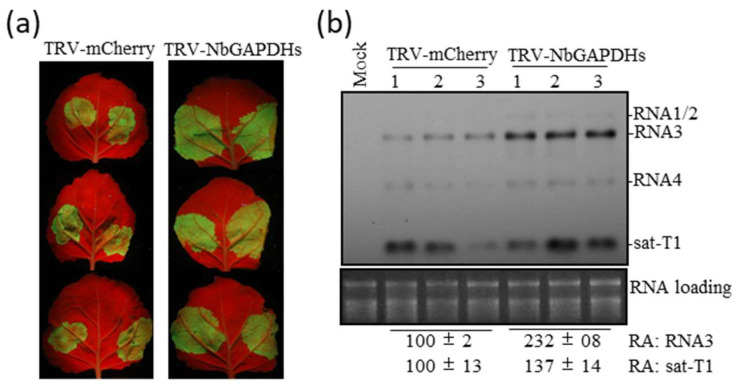
Simultaneous down-regulation of three GAPDH homologs increased the accumulation of GFP-expressing CMV variant (CMV-gfp) and sat-T1 in *N. benthamiana*. (**a**) Observation of GFP fluorescence. *N. benthamiana* plant leaves were pre-inoculated with either TRV-mCherry or TRV-NbGAPDHs via agroinfiltration. At 10 days post-infiltration (dpi), the upper expanded leaves were infiltrated with *Agrobacterium* cells for co-inoculation with CMV-gfp and sat-T1. Leaves were photographed under UV light at 3 dpi. Inoculation with TRV-GAPDHs was performed by infiltrating plants with a bacterial cell mixture harboring TRV1 with TRV2-GAPDH1, TRV2-GAPDH2, and TRV2-GAPDH3 together. (**b**) Northern blotting analyses of viral RNAs and sat-T1 in the infiltrated leaves. Total RNAs were extracted from the infiltrated leaves at 3 dpi. CMV RNAs and sat-T1 were simultaneously detected by co-incubating probes for CMV and sat-T1 in the RNA hybridization. Ribosomal RNAs were used as loading assessment for Northern blot analyses. Relative accumulation (RA) values of RNA3 and sat-T1 are shown at the bottom.

**Table 1 viruses-15-02039-t001:** γRE-binding host proteins corresponding to unique peptides identified by LC/MS-MS.

GenBank ID	Host Protein	Biological Process	Molecular Function	Cellular Component
LC015772	PDR9 mRNA for PDR-type ABC transporter	Transmembrane transport	Transporter activity	Plasma membrane
KM986324	Cytosolic glyceraldehyde-3-phosphate dehydrogenase 2 (GAPDH2)	Glycolytic process	Oxidoreductase activity	Cytosol
EF189156	Histone 2b (H2b)	Regulation of gene expression	DNA binding	Nucleus
MH532570	Nucleotide-binding leucine-rich repeat protein (ZAR1)	Defense response to bacteria	Protein kinase activity	Plasma membrane
EF661029	Histone H3 (H3)	Regulation of gene expression	DNA binding	Nucleus
LC422005	Small GTPase Ran3 (Ran3)	GTP catabolic process	GTPase activity	Nucleus
JN688263	Eukaryotic initiation factor 4A-14 (eIF4A)	Translation initiation	RNA helicase activity	Cytosol
EU862550	Ubiquitin (Ub)	Protein ubiquitination	Protein tag	Cytosol, nucleus
AF290567	Plastid lipid-associated protein 2 (PAP2)	Unknown	Unknown	Plastid, chloroplast

## Data Availability

Not applicable.

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
