# Peer review of "Identification of Host Factors Interacting with a γ-Shaped RNA Element from a Plant Virus-Associated Satellite RNA"

_viruses, 2023, doi:10.3390/v15102039_

Round 1

Reviewer 1 Report

Please see the attached comments. No big issues on this study. Just some minor issues. 

Author Response

Dear reviewer,

Thank you so much for your positive comments and corrections.  I have made all corrections and updated the reference styles to fit the requirements of the Viruses journal. I hope the revised version can be accepted for publication. 

Zhiyou

Reviewer 2 Report

Manuscript viruses-2640480 describes the identification of Nicotiana benthamiana proteins potentially interacting with the satRNA associated with cucumber mosaic virus (CMV). This research is of interest as it adds new information to an increasingly body of knowledge on CMV-host interactions. This research is thoroughly conducted. Unfortunately, the manuscript, although overall well written, would benefit from some editing. See suggestions below:

Line 8: We previously identified a highly conserved …

Lines 9-10: to be structurally required for sat RNA survival and inhibition of CMV replication. It remains …

Lines 10-11: unknown how γRE biologically functions. In this work …

Lines 12-13: … as baits. Nice host factors were …

Line 14: … via tobacco rattle virus-induced …

Lines 15 and 17: Change infections to infection

Line 15: what is sat-T1?

Line 15: change 9 and 3 to nine and three

Line 19: change regulates to regulate

Line 25: eliminate on

Line 26: This claim is incorrect because some satRNA encode proteins. Do you mean most small satRNA dot not encode function proteins?

Line 28: change sources to components

Line 29: change provides to provide

Line 33: Some plants viruses have been found …

Line 34: the name of the viruses should not be italicized

Line 47: … components of the RNA silencing machinery …

Line 60: … symptoms caused by CMV attracts aphids for …

Line 62-64: virus names should not be italicized or capitalized

Line 65: Change reduced to reduce

Line 72: Change supposed to hypothesized

Lines 77, 78, and 93: Change 9 to nine

Line 99: Italicize Agrobacterium tumefaciens

Lines 108 and 109: Change Streptavidin to streptavidin

Line 115: change Chromatographic to chromatographic

Line 116: Provide the composition of the RNA binding buffer

Line 118-119: Change collected from elution to collected by elution.

Line 143: Italicize Agrobacterium

Line 147: What do these abbreviations stand for?

Lines 150-151: What do these abbreviation stand for

Lines 157-158: tomato bushy stunt should bot be italicized

Line 203: Change help virus to HP

Line 216: Change samples (1#-3#) to samples (1-3)

Line 222: change To seek for to To determine

Line 225: change Meanwhile, to In parallel,

Line 228: Change is to was

Line 228-232: … indicating no difference between treatments and control likely due to a low abundance of protein interacting with γRE that were below the detection limit of silver staining.

Line 234: change Biotin to biotin

Line 242: Which five protein samples?

Lines 242-244: Five protein samples were subjected to LC-MS/MS. Data showed a total of 139 unique peptides identified in these samples, among which 27 were commonly present (Figure 2c)….

Line 246: … two of them corresponded to the CMV CP and the remaining matches with nice different proteins of …

Line 256: to determine how relevant these γRE-binding host proteins are, we employed …

Line 257: Eliminate technique

Line 258: Change 9 to nine

Line 269-270: Change proteins to mRNA

Line 275: Change indicate to indicated

Line 281: Eliminate to a different extent

Line 281: Change suggest to suggested

Line 282: change 4 to four

Line 283; Change is to was

Line 363: Change TRV-NbGAPC3 to TRV2-NbGAPC3

Line 388: Amongst these host proteins of interest, seven of them were evaluated for their role in their replication …

Lines 401-402: … plays a role in the replication …

Line 403: Italicize Arabidopsis

Line 428: Change promoting to leading

Line 471: Change Tobacco to tobacco

See suggested edits to improve the language

Author Response

Dear reviewer, 

Thank you so much for your comments and in-depth English corrections. We have revised the manuscript with the corrections as you suggested, which improved the manuscript very much.  The one-by-one responses to your comments are attached with a Word file.  We hope the revised version can be accepted for publication. Thanks again for your input. 

Best regards,

Zhiyou
